# Solar Light Photoactive Floating Polyaniline/TiO_2_ Composites for Water Remediation

**DOI:** 10.3390/nano11113071

**Published:** 2021-11-15

**Authors:** Ermelinda Falletta, Anna Bruni, Marta Sartirana, Daria C. Boffito, Giuseppina Cerrato, Alessia Giordana, Ridha Djellabi, Erfan S. Khatibi, Claudia L. Bianchi

**Affiliations:** 1Department of Chemistry, Università degli Studi di Milano, Via Golgi 19, 20133 Milano, Italy; anna.bruni@unimi.it (A.B.); marta.sartirana@unimi.it (M.S.); ridha.djellabi@unimi.it (R.D.); erfan.saberkhatibi@studenti.unimi.it (E.S.K.); claudia.bianchi@unimi.it (C.L.B.); 2Consorzio Interuniversitario Nazionale per la Scienza e Tecnologia dei Materiali (INSTM), Via Giusti 9, 50121 Florence, Italy; 3Polytechnique Montréal—Génie Chimique 2900 Boul, Edouard Montpetit—H3T 1J4, Montréal, QC H3C 3A7, Canada; daria-camilla.boffito@polymtl.ca; 4Department of Chemistry, Università degli Studi di Torino, Via Pietro Giuria, 7, 10125 Torino, Italy; giuseppina.cerrato@unito.it (G.C.); alessia.giordana@unito.it (A.G.)

**Keywords:** floating materials, composites, polyaniline, titania, water remediation, photocatalysis

## Abstract

In the present study, the development of innovative polyurethane-polyaniline/TiO_2_ modified floating materials applied in the sorption and photodegradation of rhodamine B from water matrix under solar light irradiation is reported. All the materials were fabricated with inexpensive and easy approaches and were properly characterized. The effect of the kind of polyaniline (PANI) dopant on the materials’ behavior was investigated, as well as the role of the conducting polymer in the pollutant abatement on the basis of its physico-chemical characteristics. Rhodamine B is removed by adsorption and/or photodegradation processes depending on the type of doping agent used for PANI protonation. The best materials were subjected to recycle tests in order to demonstrate their stability under the reaction conditions. The main transformation products formed during the photodegradation process were identified by ultraperformance liquid chromatography-mass spectrometry (UPLC/MS). The results demonstrated that photoactive floating PANI/TiO_2_ composites are useful alternatives to common powder photocatalysts for the degradation of cationic dyes.

## 1. Introduction

In the last decades, the rapid growth of the world’s population and the expansion of industrialization and agriculture have caused an exponential increase of fresh water demand. Therefore, water reuse has emerged as an important challenge, especially in regions suffering from water scarcity. Moreover, in addition to traditional pollutants, the presence of new contaminants in polluted water has become a global issue because conventional wastewater treatment processes lead to an incomplete removal of these recalcitrant compounds [1,2,3,4]. As a consequence, the scientific community has addressed its efforts to the development of highly efficient low-cost and low-impact treatments. It has been demonstrated that water pollution is responsible for 1.5 million human deaths every year [1]. Among the different aqueous pollutants, dyes have become a global issue owing to their diffusion and because they exhibit mutagenic, immunogenic, carcinogenic, and teratogenic characteristics [5,6]. Dyes find application in numerous and different fields ranging from coloring textures to plastics, passing from inks, cosmetics and so on [7]. Rhodamine B (RB) is a cationic xanthene dye used in the printing, textile, and photography industries. RB is toxic to human beings and its use in foodstuffs is prohibited in many countries; however, it is still used illegally. For this reason, we chose RB as the model molecule. Over the years, a large number of synthetic dyes have been synthesized to address diverse market demands. Even at low concentrations, the presence of dyes has a high impact on water bodies, causing an increase of the biochemical oxygen demand (BOD) level, a decrease of both the light penetration and gas solubility with negative consequences for the aquatic life, as well as for humans and animals [8].

Therefore, the discharge of wastewater containing these kinds of contaminants without proper treatments leads to serious environmental and health repercussions [9].

Numerous different technologies have been developed for treating polluted water, such as biological treatments, chemical oxidation, filtration, ozone treatments, and so forth [10,11,12].

Among these approaches, adsorption is one of the most used treatments for the removal of pollutants from the water matrix thanks to its simplicity, low cost, high efficiency, and sustainability [13,14,15,16,17]. In fact, adsorption processes based on the use of environmentally-friendly adsorption materials, such as adsorbents originating from agricultural sources and by-products (fruits, vegetables, foods), agricultural waste and derivatives, biochar, hydroxyapatite, and so forth, represent a new generation of sustainable treatments that pave the way for new scenarios in this sector. Although in the last decades, numerous different types of adsorbents have been developed (metal-organic frameworks [18], carbon nanotubes [19], graphene [20], biochar [21], activated carbon [22], etc.), this technique is defined as a mass transfer process by which a substance is transferred from the liquid phase to the surface of solid sorbents. As a consequence, once saturated, the materials must be regenerated [23,24,25]. This causes an increase in the process costs and the transfer of the pollutant to other treatments for the subsequent degradation or recovery. However, the presence of a sorbent component could be strategic to remove unreacted pollutants or recalcitrant contaminants.

Advanced oxidation processes (AOP) offer important alternative perspectives to sorption processes [2,26]. Among these approaches, photocatalytic degradation has a special place, being able to degrade many pollutants without the addition of chemical oxidants under mild operating conditions [2,27,28,29,30]. In recent years, the attention of the scientific community towards photodegradation processes under sunlight has gradually grown [31,32,33], as have investigations of recyclability and regeneration tests on photoactive materials [34,35,36].

TiO_2_ is still one of the most investigated semiconductors for photodegradation reactions, thanks to its chemical stability, wide availability, and low cost [14,37,38]. However, even though it is highly active under UV light irradiation, its activity under sunlight is very low.

In recent years, conducting polymers (CPs)-based photocatalysts have been tested for the photodegradation of different types of water pollutants [37,38,39]. In fact, it has been demonstrated that CPs can act as excellent photosensitizers thanks to their extended π-configuration system, which extends the TiO_2_ activity to the visible region. Under visible light irradiation, in CPs/TiO_2_ composites, photoelectrons are injected into the conduction band of TiO_2_, whereas the photoholes at the Highest Occupied Molecular Orbit (HOMO) of CPs lead to the oxidation of water to hydroxyl radicals. These processes lead to the formation of peroxides and hydroxyls that, further, take part in the mineralization of organic species. It is also important to note that CPs are good adsorbents for the removal of not only small organic molecules, which can escape the photodegradation process, but also dangerous heavy metals [14,15,16,17]. Among the CPs, polyaniline (PANI) is unique thanks to its ease of synthesis, environmental stability, redox properties and unique doping/dedoping process [40,41].

Among the different approaches proposed so far for PANI synthesis, chemical polymerization is the oldest and still the most popular way thanks to its affordability and simplicity [42,43,44,45,46,47,48,49,50].

For applications with sunlight as the irradiation source, the development of water-floating photocatalysts could be one of the best options due to their characteristics in terms of efficiency, thanks to a high oxygenation of the photocatalyst’s surface, a full sunlight irradiation, easy recovery, and reuse.

In the present work, the synthesis of innovative easily recoverable PANI-TiO_2_ composites supported on a commercial low-cost polyurethane (PU) floating foam (PT/PU) is described, as well as their ability to remove rhodamine B (RHB, model dye) from the water matrix. We describe the effect of the type of PANI dopant on the properties of the material, which is able to enhance its adsorbing or photocatalytic properties. Finally, the transformation products (TPs) deriving from the photodegradation process were properly identified.

## 2. Materials and Methods

### 2.1. Materials

For the preparation of all the materials, Merck reagents (Darmstadt, Germany) and ultrapure (UP) water were used. All the reagents were of analytical purity and were used as received. Titanium dioxide nanopowder P-25 was supplied by Evonik (Bitterfeld, Germany). A commercial polyurethane (PU) reticulated foam (Amtra pro nature, Rodgau, Germany) was used as materials support. PU was used in the form of cubes 3 cm × 3 cm × 1 cm (0.2 g). For both HPLC/UV and UPLC/MS analyses, HPLC-grade acetonitrile and water were purchased from VWR Chemicals (Darmstadt, Germany) and Carlo Erba Reagents S. r. l. (Cornaredo, Milan, Italy), respectively.

### 2.2. Synthesis of the Materials

PT/PU materials were prepared by different approaches, as described below.

#### 2.2.1. In Situ Polymerization (P/PU and PT/PU-1 Samples)

For the preparation of P/PU, 1 mL of aniline was dissolved in 90 mL of 1 M HCl. The solution was stirred for 1 h in an ice bath in the presence of a PU cube. Then, 100 mL of 0.1 M ammonium persulphate (APS) solution (2.5 g APS in 100 of 1 M HCl) was added drop by drop into the previous mixture. After 4 h, the reaction was stopped by the addition of 10 mL of acetone. The modified PU cube was recovered and washed several times with deionized water and acetone and dried in air.

PT/PU-1 was prepared by the same procedure used for P/PU but adding 0.5 g of TiO_2_ NPs into the aniline chloride solution before the addition of the PU cube.

#### 2.2.2. Immobilization (PT/PU-2 Sample)

A cube of P/PU was immersed in a water suspension containing 1.5 g of TiO_2_ under stirring. After 30 min, the foam was recovered, abundantly washed with deionized water and dried in air.

Any attempt to immobilize pristine TiO_2_ NPs on PU failed.

#### 2.2.3. In Situ UV-Assisted Polymerization (PT/PU-3)

For the preparation of PT/PU-3 1 mL of aniline was solubilized in 200 mL 1 M HCl. After 30 min, a PU cube was introduced, followed by 100 mg TiO_2_ NPs. After a further 30 min, the mixture was UV irradiated for 3 h. Then, 10 mL of H_2_O_2_ (30% *w/w*) was added along with 0.5 mL of an Fe(III) solution 4 mg/mL. The reaction was stopped by the addition of 10 mL of acetone. The modified PU cube was recovered and washed several times with deionized water and acetone and dried in air.

#### 2.2.4. Impregnation by Organic Dopants-Modified PANIs Solutions (PT/PU-4 and -5)

Two solutions of camphorsulfonic acid (CSA)- and dodecylbenzenesulfonic acid (DBSA)-doped PANIs in chloroform were prepared as described elsewhere [39,40]. Briefly, 1 g of PANI (the green powder recovered during the preparation of P/PU) was undoped (deprotonated) by 100 mL of 1 M NH_4_OH. After 24 h the product was filtered, washed several times with water until the mother liquid became neutral and dried in air. Then, 0.5 g of deprotonated PANI was suspended in 100 mL of chloroform and 880 mg of dodecylbenzenesulfonic acid (DBSA, aniline/DBSA = 2 molar ratio) or 630 mg of camphorsulfonic acid (CSA, aniline/CSA = 2 molar ratio) was properly added. Each mixture was stirred for 24 h at room temperature and then filtered. The solutions were concentrated, reaching a concentration of about 20 mg/mL.

TiO_2_ NPs (0.4 g) were dispersed in 5 mL of each organic acid-doped PANI solution. The mixture was stirred for 10 min. Afterwards, a PU cube was added to each mixture and impregnated for 1 min. The obtained modified PU foams (DBSA-doped PT/PU, PT/PU-4 and CSA-doped PT/PU, PT/PU-5) were recovered and dried in air.

### 2.3. Characterization

The morphologic characterization was carried out using a scanning electron microscope operating with a Field Emission source (model TESCAN S9000G; Source: Schottky type FEG; Resolution: 0.7 nm at 15 keV (in In-Beam SE mode, Brno – Kohoutovice, Czech Republic)), without any pre-treatment of the samples.

The ATR-FTIR spectra of the samples were recorded using a Bruker Vertex 70 spectrophotometer (Bruker, Billerica, MA, US) equipped with the Harrick MVP2 ATR cell (resolution 4 cm^−1^).

Contact angles were measured with 5 μL of ultrapure water droplets by means of an EasyDrop Contact Angle Measuring Instrument (KRUSS GmbH, Hamburg, Germany) at room temperature. All the contact angles were determined by averaging the values obtained at two to three different points on each sample surface.

For the UV-vis analyses, a T60 UV-visible Spectrophotometer PRIXMA (PG instruments, Lutterworth, United Kingdom) was used, operating in a wavelength range of 450–650 nm.

Chromatographic separations were carried out by an HPLC Agilent 1100 Series (Santa Clara, CA, USA) equipped with UV-vis detector and a C18 Supelco column (Darmstadt, Germany) (25 cm × 4 mm, 5 µm).

For monitoring RHB depletion and TPs formation during photodegradations tests, a UPLC Ultimate 3000 (Thermo Fisher Scientific, Waltham, MA, USA), equipped with an autosampler, temperature-controlled column compartment and UV detector, was used, interfaced with a Thermo Fisher LCQ Fleet ion trap mass spectrometer (MS) with an electrospray ionization source (ESI) (Thermo Scientific, Waltham, MA, USA).

### 2.4. Rhodamine B Abatement Tests

The experiments for RHB abatement were performed using a 250 mL batch glass reactor filled with 100 mL of a 10 ppm RHB solution in ultrapure water at spontaneous pH (about 5). The P/PU or PT/PU foams were maintained under constant magnetic stirring to ensure the mixing of the solution. Before irradiation, each suspension was stirred for 30 min in the dark, followed by 180 min under solar light irradiation. The light source, placed above the reactor, was a 300 W commercial solar lamp (ULTRA VITALUX 300 W-OSRAM, OSRAM, Múnich, Germany) with an effective irradiance of 35 W∙m^−2^. The RHB abatement was monitored for 210 min, withdrawing aliquots every 15 min in the first half-hour and every 30 min in the subsequent three h. All the solutions were analyzed by UV-vis spectroscopy (λ = 554 nm) and HPLC chromatography. Chromatographic separations were carried out using an isocratic elution at a flow rate of 1.00 mL∙min^−1^. The mobile phase was composed of 0.1% formic acid in water (65%) and acetonitrile (35%). The run time was 35 min. The injection volume was 20 µL and the detection wavelength 554 nm.

The same samples were analyzed with the UPLC-MS technique using the same chromatographic conditions in order to monitor and identify the RHB TPs during the photodegradation tests. The MS interface conditions for sample acquisition were the following: heater temperature 150 °C, sheath gas flow rate (arb) 20, auxiliary gas flow rate (arb) 10, sweep gas flow rate (arb) 10, spray voltage negative mode 3.50 kV, capillary temperature 275 °C, capillary voltage −10 V, tube lens −10 V, m/z range 50–500 Da.

After the first abatement test, the coated-PU foams that showed the best performances were rinsed with water or acetone and used for further cycles.

## 3. Results

### 3.1. Characterization of the PT/PU Foams

The materials that showed the best performances were characterized by different techniques, as reported below.

ATR-FTIR spectroscopy. Figure 1 displays the ATR-FTIR spectra of TiO_2_ powder, fresh and used PT/PU-2 and fresh and used PT/PU-3, respectively.

Since fresh and used PT/PU-3 samples show ATR-FTIR spectra that are very similar to that of the PU support, the differential spectra were obtained (in Figure 1D), which allow us to highlight the signals of the conducting polymer.

In particular, the band at 1500 cm^−1^ is assigned to benzene rings of PANI, while that at 1600 cm^−1^ to quinone rings [14,15,16,17,51]. For all the PANI-modified PUs, no signals specifically attributable to titania are recognized, since in the ATR-FTIR spectrum of PANI many bands below 900 cm^−1^ are observed (Appendix A). Moreover, the synthetic method employed for aniline polymerization does not seem to have strongly affected the structural characteristics of the polymer. In more detail, the spectra of the PT/PU samples can be considered the sum of the signals of the coating polymer and the support (PU).

Comparing the ATR-FTIR spectra of samples before and after use, it is possible to notice in general a decrease of the band assigned to benzene rings, suggesting that the oxidative process had partially modified the material. The results of oxidative processes are more evident for the PT/PU-2 sample before and after use: in the spectrum of used material, two new bands appear at 1340 and 855 cm^−1^, respectively (indicated with a star in the red curve in Figure 1B), suggesting that the polymer backbone could be modified by the oxidative process. In particular, these signals may be attributable to the presence of semiquinone radical cations formed on the material’s surface [52].

The difference in the oxidation level of the PANI backbone is strictly related to the kind of oxidant used. In fact, for the preparation of PT/PU-2, a traditional strong oxidant was employed (APS) that guarantees both the aniline polymerization and the oxidation process of the polymer chains. On the other hand, the UV-assisted synthesis leading to PT/PU-3 was carried out by means of hydrogen peroxide (H_2_O_2_) as oxidant and Fe^3+^ as catalyst. As reported in the literature [53], with this approach, the oxidation level of the polymeric chains, as well as other properties, can be influenced by several parameters such as reaction time, aniline/oxidant ratio, and so forth.

ATR-FTIR spectra of PT/PU-4 e PT-PU/5 show a large number of signals, due to the presence of organic dopants and, as a consequence, the band assignment is complex. It is interesting to observe that both the samples present a sharp signal at 3020 cm^−1^, absent in the spectra of inorganic-doped samples, indicated with a star in Figure 1E. This signal could be attributable to stretching of the CH bond, probably shifted for the electron-withdrawing effect of sulfonic groups.

Morphology. Figure 2 shows the FESEM images and the EDS analyses of fresh and used PT/PU-3.

Both the foams have a relatively homogeneous and regular globular or pseudo-globular morphology, with an approximate size of the order of 50 nm or less. The presence of titania is confirmed by the EDX survey, in which it is possible to identify a certain localization of the titanium. The UV treatment does not seem to affect the texture of the PANI polymer matrix.

For organic-doped foams (PT/PU-4 and -5), similar results were obtained (Appendix A).

Water Contact Angle (WCA) Measurements. To have information on the interactions at the solid–vapor, solid–liquid, and liquid–vapor interfaces, the measure of the contact angle (CA), θ, of a liquid drop on a solid surface is particularly useful [54]. It was demonstrated that the wettability of a material is determined by its surface tension [55]. Although many parameters, such as temperature [56], light [57], roughness [58], and surface morphology [59] can affect CA measurements, typically, materials characterized by low wettability show high CA values, whereas those with high wettability exhibit low CA values. For the application in the field of water remediation, the hydrophilic properties of the sorbent or photoactive material can play a crucial role, allowing full contact with hydrophilic pollutants.

Table 1 summarizes the measures of WCA obtained for all the modified foams, whereas Appendix A reports the corresponding digital images.

As reported in the literature, PANI is a relatively hydrophobic polymer showing values of WCA between 78 and 44° depending on the protonation level [60]. However, the type of doping agent employed can affect its hydrophilicity [61,62].

As it is possible to observe from the data reported in Table 1, all the HCl-doped PT/PUs maintain values of hydrophobicity similar to P/PU. However, replacing HCl with the organic doping agents (DBSA and CSA), it is possible to tune the hydrophilicity of the material. In fact, the long non-polar chain of DBSA increases the hydrophobicity of the foam, whereas the oxo group of CSA takes part in hydrogen bonding, making PT/PU foams more hydrophilic.

### 3.2. Assessment of RHB Abatement Capacity

It has been demonstrated that PANIs are good candidates for the removal of dyes from the water matrix playing as sorbents, both as powder [14,16,17] and properly supported on different materials [59].

Several synthetic parameters can affect the material’s properties, such as type of oxidant used, synthetic strategy, kind of doping agent, and so forth. Therefore, in the present work, the materials were classified into two main categories: HCl-doped PT/PUs and DBSA- or CSA-doped PT/PUs.

#### 3.2.1. RHB Abatement Capacity of HCl-Doped PT/PUs

Figure 3 displays the results obtained for RHB abatement by means of P/PU in the absence of light irradiation and PT/PU-1, PT/PU-2 and PT/PU-3 under light irradiation.

P/PU has low affinity towards RHB. The low percentage of adsorption (about 15%) can be attributed to the presence of positive charges on the surface of the material (due to the HCl-doped imine groups of PANI) that do not promote cationic dyes adsorption. In fact, PANI-based materials are promising sorbents for anionic dyes [14,16,17]. Their main sorption mechanism is based on an anion-exchange process (dopant anion vs dye), even if short-range interactions, such as hydrogen bonding and π-π stacking, can occur [17,63,64,65]. Concerning the TiO_2_ loaded materials, if on the one hand PT/PU-1 exhibits poor photoactive properties, on the other hand PT/PU-2 and -3 show very promising performances, reaching more than 70% of RHB degradation after 180 min of light irradiation.

As reported in the literature, PANI is capable of photosensitizing TiO_2_ efficiently [27], leading to the production of a large number of electron-hole pairs under visible light illumination, resulting in higher photocatalytic activity than pristine TiO_2_. In fact, it has been demonstrated that PANI modification provides an effective approach to extending the TiO_2_ absorption to the visible light range, predominantly present in sunlight.

However, the results shown in Figure 3 suggest that the synthetic approach employed for the preparation of PANI/TiO_2_ modified-PU plays a key role in the performances of the final materials.

Even though it is hard to evaluate, TiO_2_ dispersion and its load on the polymeric matrix could be responsible of the different behaviors of the tested foams. This is confirmed by the fact that, for both PT/PU-1 and -2, the polyaniline coating was carried out by the same synthetic approach, whereas the TiO_2_ loading was performed in two different ways: in situ during the oxidative polymerization of aniline for PT/PU-1 and by subsequent impregnation of PT/PU-2.

An important parameter that defines the functionality of sorbent or photoactive material is its capacity to be regenerated and reused in several cycles. For this reason, the PT/PU materials that showed the best performances (PT/PU-2 and -3) were subjected to recycle tests after washing with water. Appendix A shows the drastic decrease in activity for both the materials. According to the literature [66], the fall of the photodegradation efficiency of the foams could be mainly attributed to the strong adsorption of the dye intermediates and products on the surface of the photocatalyst (mainly on the polymeric matrix) leading to the blockage of photocatalytic active sites (PANI/TiO_2_) [27], whereas losses of photocatalyst from the surface of the foam during the consecutive tests can be excluded.

In order to desorb these compounds from the PANI matrix efficiently, and recover at least part of the properties of the materials, washing with an organic solvent (acetone) is necessary, as displayed in Figure 4.

Even though, on the one hand, a slight coloring of the washing solution demonstrated that some of the products have been desorbed, on the other hand, evidently, the desorption process is partial, since after four cycles both the materials lose any activity.

Despite the improvements, especially for PT/PU-2 sample, the photoactivity of both materials is completely suppressed within four cycles.

#### 3.2.2. RHB Abatement Capacity of Organic Acids-Doped PT/PUs

On the basis of previous investigations of the dependence of PANI performances from the composition of the material, in particular from the kind of doping agent used [14], other floating photoactive PANI/TiO_2_ based materials were prepared, changing the type of doping agent employed, and the results are reported in Figure 5.

The surprising results show the remarkable effect of the kind of doping agent on the activity of the materials. The long aliphatic chain present in the structure of DBSA converts the floating photocatalysts in a highly efficient adsorbent but totally suppresses its photoactivity. In fact, after 30 min of exposure in the dark, the dye removal is practically complete. As confirmed by the WCA measurements (Table 1), the aliphatic chain of the dopant makes the foam hydrophobic, not allowing the interaction between the polar sites of the dye molecules and the photoactive sites (PANI/TiO_2_), but promoting RHB-DBSA hydrophobic interactions. On the contrary, the smaller steric hindrance and the high hydrophilicity of CSA guarantee the interaction between the pollutant molecules and the photoactive component of the materials leading to a high RHB abatement after 150 min of solar light irradiation.

After the first test, each material was quickly washed with ultrapure water and was reused for a further two cycles, as depicted in Figure 6.

DBSA-doped material (PT/PU-4) shows a gradual decrease of their adsorbent capability within the three performed tests, passing from the ca. 100% of RHB adsorption of the first test to the ca. 86% of the third run, because of the partial saturation of the polymeric component, where irradiation might not play a significant role.

Concerning the CSA-doped material (PT/PU-5), after the first run, where a RHB photodegradation of ca. 100% is obtained, a decrease in RHB degradation rate is observed leading to a ca. 70% of RHB degradation in 180 min, which, however, remains constant during the third test. It is possible to affirm that, unlike HCl, dopant CSA contributes to preventing the saturation of active sites. In fact, the high hydrophilicity of the foam surface permits a continuous washing of the material. The TPs resulting from the RHB solution treated by means of PT/PU-5 during the first test were properly identified as described below.

### 3.3. Identification of Transformation Products

The identification of TPs, resulting from the RHB solution treated by means of PT/PU-5 during the first test, was carried out by UPLC/ESI-MS in positive mode. During the photodegradation under solar light irradiation, two competitive processes can occur simultaneously—N-deethylation and destruction of dye chromophore structure [38,67]. In the RHB solution treated with PT/PU-5, TPs deriving from both the degradation processes were identified, along with a new byproduct that had an *m/z* 231 as reported in Figure 7.

More polar TPs observed in the first part of the LC chromatogram can be attributed to the formation of small organic acids, whose low-molecular weight (e.g., benzoic acid, pyrocatechol, glutaric acid, phthalic acid etc.) requires the use of a different analytical technique, such as GC/MS, for detection and identification [68].

If on the one hand these results clearly confirm the photodegradation properties of the synthesized PT/PU materials, on the other hand, the incomplete degradation of rhodamine B leads to the persistence in solution of TPs whose toxicity must be carefully evaluated.

## 4. Conclusions

Efficient photoactive floating PANI/TiO_2_ composites were proposed in this work. It was demonstrated that the key role of the PANI doping agent was in tuning both the hydrophilicity and the photoactivity of the materials. In more detail, the use of an inorganic dopant (HCl) leads to photoactive material that is hard to regenerate after the first test. The replacement of the inorganic dopant with organic acids (CSA and DBSA) allows the fine control of the surface properties of the materials. The long nonpolar chain of DBSA increases the hydrophobicity of the foam, making the material a good adsorbent towards RHB removal (100% RHB abatement) but suppressing at the same time its photoactivity, not allowing the interaction between the polar sites of the dye molecule and the photoactive sites (PANI/TiO_2_). On the contrary, CSA-doped PT/PU showed exciting results leading to 100% of RHB photodegradation during the first run and maintaining about 80% of RHB conversion during the subsequent photodegradation tests. In fact, the higher hydrophilicity of this material allows, on the one hand, a strict interaction between the pollutant and the photocatalyst and, on the other hand, the continuous cleaning of the surface.

## Figures and Tables

**Figure 1 nanomaterials-11-03071-f001:**
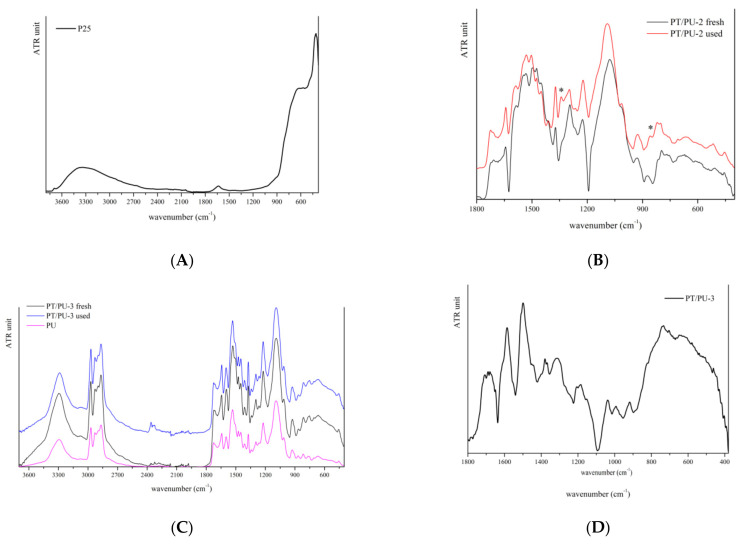
ATRFTIR spectra of (**A**) TiO_2_ powder, (**B**) fresh and used PT/PU-2, (**C**) fresh and used PT/PU-3, (**D**) PT/PU-3 differential spectrum, (**E**) PT/PU-4 and -5.

**Figure 2 nanomaterials-11-03071-f002:**
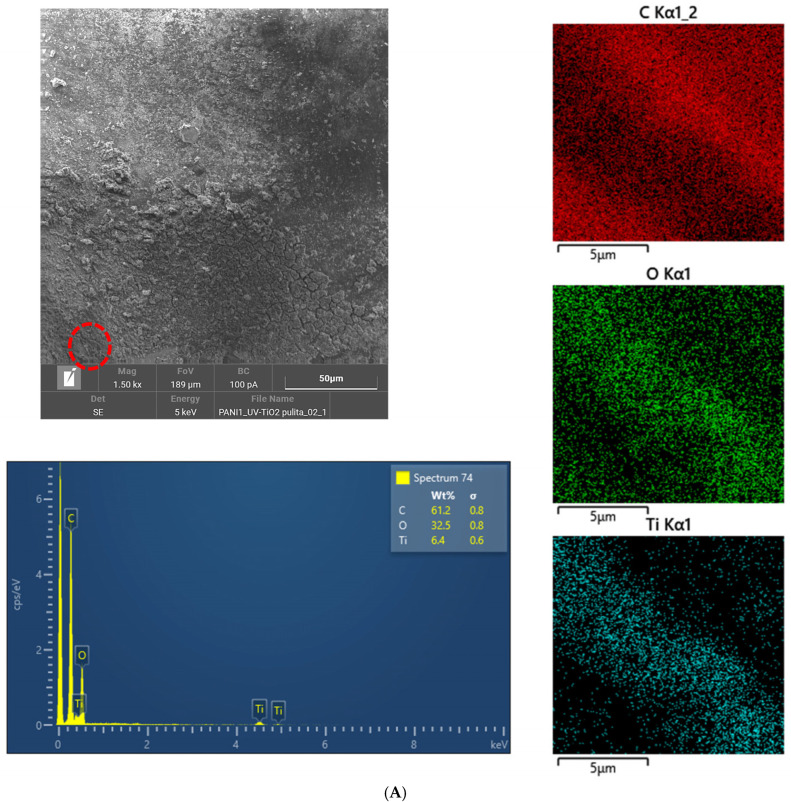
FESEM images and EDS analyses of both (**A**) fresh and (**B**) used PT/PU-3.

**Figure 3 nanomaterials-11-03071-f003:**
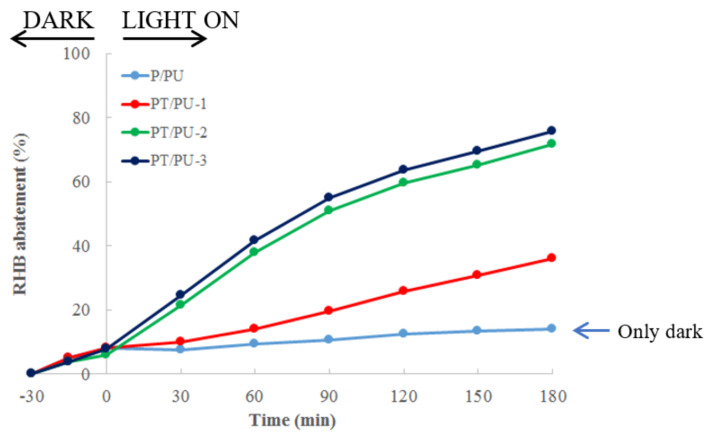
Percentage of RHB abatement tests using P/PU (only in the dark), PT/PU-1, PT/PU-2 and PT/PU-3 (30 min in the dark and 180 min under solar light irradiation).

**Figure 4 nanomaterials-11-03071-f004:**
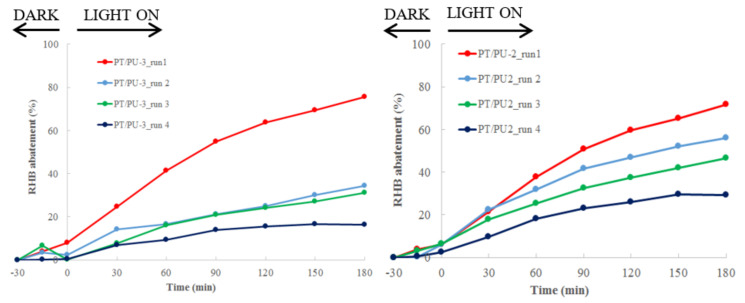
Reuse of PT/PU-2 (**left**) and -3 (**right**) for RHB abatement after washing with acetone.

**Figure 5 nanomaterials-11-03071-f005:**
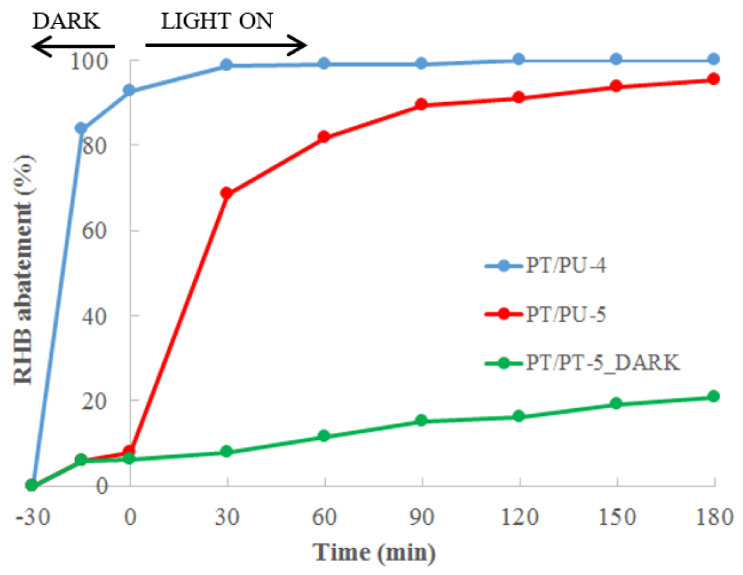
Percentage of RHB abatement tests using PT/PU-4 and -5.

**Figure 6 nanomaterials-11-03071-f006:**
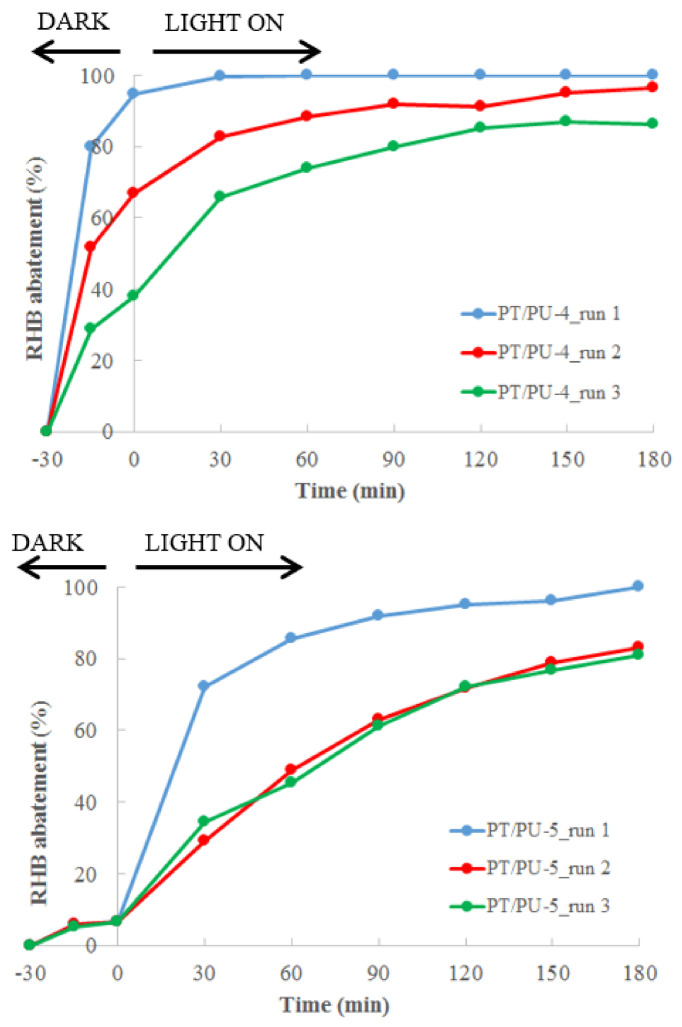
Reuse of PT/PU-4 (**up**) and -5 (**down**) for RHB abatement after washing with UP water.

**Figure 7 nanomaterials-11-03071-f007:**
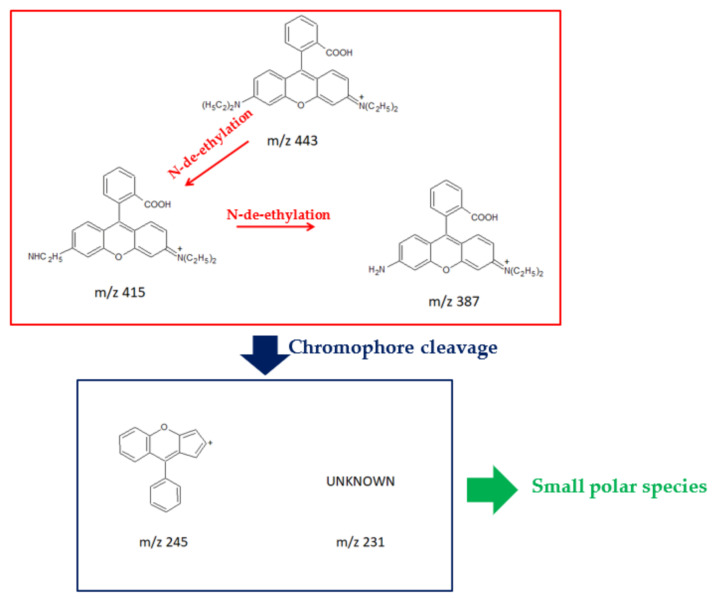
Identified RHB TPs.

**Table 1 nanomaterials-11-03071-t001:** Equilibrium values of contact angle for the synthesized P/PU and PT/PUs.

SAMPLE	WCA (°)
P/PU	75.30 ± 0.50
PT/PU-1	75.70 ± 1.74
PT/PU-2	81.19 ± 1.23
PT/PU-3	89.97 ± 0.55
PT/PU-4	118.56 ± 0.00
PT/PU-5	29.40 ± 0.00

## Data Availability

Data is contained within the article or Appendix A.

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
