# Peer review of "Solar Light Photoactive Floating Polyaniline/TiO2 Composites for Water Remediation"

_nanomaterials, 2021, doi:10.3390/nano11113071_

Round 1
Reviewer 1 Report
In the present study, the authors report the development of polyurethane-polyaniline/TiO2 modified floating materials applied in the sorption and photodegradation of rhodamine B from water matrix under solar light irradiation.
The authors demonstrated that photoactive floating PANI/TiO2 composites are useful alternatives to common powder photocatalysts for the degradation of cationic dyes.
This is a quite interesting work; Nevertheless some minor revisions are needed in order to publish this manuscript.
- As the authors state .."Dyes find application in numerous and different fields ranging from coloring textures to plastics, passing from inks, cosmetics and so on.." Could the authors mention a couple of arguments why they have chosen the specific cationic dye (Rhodamine B) in this work?
- The authors used a a Bruker Vertex 70 spectrometer to collect the ATR spectra of the samples. Since the measurements were not under vacuum, have the authors purged the spectrometer with dry air, nitrogen etc.? This way there is no contribution from humidity or CO@ from the atmosphere..
- In lines 351-357 the authors should analyze further their findings regarding affect of doping on the hydrophilicity.
- As shown in this manuscript, the photoactivity of both materials is suppressed within 4 cycles. Could the authors comment a little more about that? What happens typically in the literature? Could the authors compare their findings with other works?
- Actually what the author can see is the decolorization od Rhodamine B, not its degradation. Could they the used terms to decolorization?
- There are a few typos and syntax errors that need to be resolved.
This work can be published after covering the above minor issues.
Author Response
Please, see the attachment.

Reviewer 2 Report
The paper submitted by E. Falletta and coworkers reports on the preparation of a novel material based on a polyurethane/polyaniline floating material doped with TiO2 and its use in photodegradation of Rh-B model dye under solar light irradiation. It is well known that TiO2 offers many advantages when considering its application as photocatalyst: it is plentiful, cheap, non-toxic, and chemically inert. However, while highly active under UV irradiation, its catalytic performance under visible light is low to mediocre at the best, but this can be improved when combined with a conducting polymer.
The Introduction part reads well and provides an adequate review of the research problem and solutions provided so far. The synthesis is quite simple and described in detail. The characterization was performed by standard methods like SEM/EDX, FTIR (ATR), and Contact angle measurement. The photodegradation of Rh-B was followed using UV/VIS, HPLC and UPLC-MS techniques.
The study seems to be concise and should be of interest within the community researching the area of nanomaterials and especially photocatalysis. The research design seems to be appropriate and the used characterization methods are in my opinion sufficient to support the conclusions.
As to conclude, I recommend publishing the submitted paper after some minor issues have been corrected:
General remark: while the English language used throughout the manuscript is mostly appropriate, there are some minor shortcomings:
Abstract, lines 16 – 18:
In the present study, it is reported the development of innovative polyurethane-polyaniline/TiO2 modified floating materials applied in the sorption and photodegradation of rhodamine from water matrix under solar light irradiation.
A different order of words would sound better, e.g.:
In the present study, the development of innovative polyurethane-polyaniline/TiO2 modified floating materials applied in the sorption and photodegradation of rhodamine from water matrix under solar light irradiation is reported.
Abstract, lines 21 – 23:
Rhodamine B is removed by adsorption and/or photodegradation processes depending on the type doping agent used for PANI protonation.
Should read:
Rhodamine B is removed by adsorption and/or photodegradation processes depending on the type of doping agent used for PANI protonation.
Introduction, lines 34 – 37:
Moreover, in addition to traditional pollutants the presence of new contaminants in polluted water have become a global issue because conventional wastewater treatment processes lead to an incomplete removal of them [2- 4].
I have some issue with this sentence. In any case, it should read:
Moreover, in addition to traditional pollutants, the presence of new contaminants in polluted water has become a global issue because conventional wastewater treatment processes lead to an incomplete removal of them [2- 4].
The last word of the sentence (‘them’) needs rewriting because it is not completely clear to what of the many nouns used in the previous sentence it refers to: traditional pollutants, new contaminants, or conventional wastewater treatment processes?
In addition, the term ‘dried at air’ is used more than once. ‘Dried in air’ would be more appropriate.
Some other minor remarks:
- Introduction, lines 53 – 55. The author claim that adsorption provides ‘simplicity, low cost, high efficiency and low impact’. The last clam, i.e. low impact, should be explained in more detail.
- Introduction, line 60: what is meant with ‘transfer of the pollutant to other treatments’ ?
- Introduction, line 65: while discussing photodegradation under UV, the authors use the term ‘ambient conditions’. I don’t believe that artificial UV light can be referred to as ‘ambient conditions’, even when performed at room temperature, normal pressure and ambient atmosphere.
- The authors often use the term ATR spectroscopy. However, ATR (attenuated total reflection) is a special sub-technique of IR/FTIR and not a stand-alone method, so I suggest using the term ATR-FTIR.
- One final remark, which has to be discussed in 3.3.: The authors state in Introduction how dyes have become a global issue due to a lot of harmful characteristics like ‘mutagenic, immunogenic, carcinogenic, and teratogenic characteristics'. In their study, the authors clearly show how their photocatalyst can be used for photodegradation of RhB into smaller molecules with m/z = 415, 387, 245 and (unidentified) 231. However, those degradation products can also have harmful characteristics, maybe even more then (fairly benign) RhB itself. While it is nice to see how the dye decomposes and the color fades, there is absolutely no warranty that the nice-looking wastewater after degradation of dyes is really less harmful when released into the environment.
Author Response
Please, see the attachment.

Reviewer 3 Report
Review of the article “Solar light photoactive floating polyaniline/TiO2 composites for water remediation” by E. Falletta, A. Bruni, M. Sartirana, D. C. Boffito, G. Cerrato, A. Giordana, R. Djellabi, E. S. Khatibi, C. L. Bianchi, Nanomaterials, Manuscript ID nanomaterials-1443545.
In this article, a polymer nanocomposite composed of TiO2 P25 nanoparticles, polyurethane and polyaniline is proposed as an innovative material for the photocatalytic degradation of wastewater pollutants under simulated solar light irradiation. The polyaniline component is prepared via in situ and UV-assisted in situ polymerization on polyurethane foam cubes, using different dopants (HCl, dodecylbenzenesulfonic acid (DBSA), camphorsulfonic acid (CSA)) for the modification of the adsorption and photocatalytic properties of the final nanocomposite. Blending and impregnation are used to immobilize TiO2 nanoparticles. ATR-FTIR and EDX measurements confirm the presence of the three components, while FESEM images reveal a 50 nm globular morphology. For the photocatalytic degradation experiments, a 1 mg/L aqueous solution of Rhodamine B (RhB) is chosen. The results reveal a low (< 10%) adsorption capacity of HCl- and CSA-doped nanocomposites, while DSBA-doped ones adsorb RhB efficiently (> 90%) in 30 minutes, thanks to the hydrophobic interactions between RhB and DSBA chains. As for the photodegradation ability, the HCl-doped samples with impregnated and UV-assisted blended TiO2 show good RhB degradation performance (> 70%) after 180 min irradiation. The photodegradation ability of DBSA-doped nanocomposites is negligible, while CSA-doped ones present the highest degradation efficiency (> 90%), due to the interactions between RhB and the polyaniline-TiO2 sites. Recyclability tests reveal that HCl-doped nanocomposites undergo a strong decrease in photocatalytic activity after 4 cycles (reduced to ca. 25%), while CSA-doped ones keep a good performance after 3 cycles (ca. 80%).
Major comments:
The nanocomposites proposed by the authors are extremely interesting, since adsorption ability and photocatalytic activity are coupled in one multifunctional material. Nevertheless, some issues about the data obtained have to be addressed by the authors:
- Are these nanocomposites mostly submerged in water? Which percentage of the surface area of the samples is possible to utilize during the experiments?
- Lines 66-70: these two statements are not exact. There is a plethora of reports on visible light and solar light activatable photocatalysts. See for example: 1) Asahi, R., Morikawa, T., Ohwaki, T., Aoki, K. & Taga, Y. Visible-Light Photocatalysis in Nitrogen-Doped Titanium Oxides. Science 293, 269-271, doi:doi:10.1126/science.1061051 (2001). 2) Chen, X., Liu, L., Yu, P. Y. & Mao, S. S. Increasing Solar Absorption for Photocatalysis with Black Hydrogenated Titanium Dioxide Nanocrystals. Science 331, 746-750, doi:doi:10.1126/science.1200448 (2011). 3) Ong, W.-J., Tan, L.-L., Ng, Y. H., Yong, S.-T. & Chai, S.-P. Graphitic Carbon Nitride (g-C3N4)-Based Photocatalysts for Artificial Photosynthesis and Environmental Remediation: Are We a Step Closer To Achieving Sustainability? Chem. Rev. 116, 7159-7329, doi:10.1021/acs.chemrev.6b00075 (2016).
Moreover, recyclability tests are routine experiments in photocatalysis. See for example: 1) Li, H. et al. Synthesis and characterization of g-C3N4/Bi2MoO6 heterojunctions with enhanced visible light photocatalytic activity. Appl. Catal. B-Environ. 160-161, 89-97, doi:https://doi.org/10.1016/j.apcatb.2014.05.019 (2014). 2) Han, C., Chen, Z., Zhang, N., Colmenares, J. C. & Xu, Y.-J. Hierarchically CdS Decorated 1D ZnO Nanorods-2D Graphene Hybrids: Low Temperature Synthesis and Enhanced Photocatalytic Performance. Adv. Funct. Mater. 25, 221-229, doi:https://doi.org/10.1002/adfm.201402443 (2015). 3) Yang, Y. et al. Preparation and enhanced visible-light photocatalytic activity of silver deposited graphitic carbon nitride plasmonic photocatalyst. Appl. Catal. B-Environ. 142-143, 828-837, doi:https://doi.org/10.1016/j.apcatb.2013.06.026 (2013).
The authors should therefore modify this paragraph.
- Lines 110-145 and 425: did the author perform leaching tests to verify that PANI, TiO2 nanoparticles, DBSA and CSA are not drained away from PU cubes with use? Which kind of interactions exist between TiO2 nanoparticles and PANI?
- Line 240: why can the signals below 900 cm-1 in the ATR-FTIR spectra not be attributed to Ti-O stretching and Ti-O-Ti bridging stretching modes?
- Line 247: for the sake of clarity, the authors should indicate explicitly the relevant graph(s).
- Line 251 and 264: please add a symbol in the ATR spectra to indicate the peaks at 855, 1340 and 3020 cm-1.
- Line 322: the image does not seem to correspond to what is stated in the text. Sample A does not contain titania. Are “fresh and used” referred to the samples before and after the photodegradation experiments?
- Lines 427-428: the authors should explain why the washing with acetone allows to recover the photocatalytic activity of these nanocomposites.
- Line 526: in Figure 6, upper graph (Reuse of PT/PU-4), it seems that the adsorption-desorption equilibrium was not reached in these experiments (especially in run 2 and 3) before the irradiation. According to the negligible photocatalytic activity of PT/PU-4, the trends shown in the graph are most likely attributable to a slower adsorption process, where irradiation might not play a significant role.
Minor comments:
- As for the ATR spectra, the authors should use higher resolution images.
- Some necessary commas are missing all over the text, such as in line 34 “in addition to traditional pollutants,”, line 90 “For applications with sunlight as the irradiation source,” and so on.
- Line 21: I suggest to use “physico-chemical” instead of “chemical-physical”.
- Line 44: please use “address” instead of “face up to”.
- Lines 78 and 79: the authors should use “photoelectrons” and “photoholes” and delete “which were obtained as a result of absorption of photons”.
- Line 94: subscript in “TiO2”.
- Line 143: please use “Afterwards,” instead of “Then”.
- Line 166: please add “B” to “Rhodamine”.
- Line 173: please use “irradiance” instead of “power density of irradiation”.
- Line 178: please delete “9” after “39%” and add “The” before “run time”.
- Line 245: superscript in “cm-1”.
- Line 341: please write “images”.
- Line 396: please write “does not”.
- Line 545: it seems that the subject of the sentence is missing.
Author Response
Please, see the attachment.

Round 2
Reviewer 3 Report
Review of the article “Solar light photoactive floating polyaniline/TiO2 composites for water remediation” by E. Falletta, A. Bruni, M. Sartirana, D. C. Boffito, G. Cerrato, A. Giordana, R. Djellabi, E. S. Khatibi, C. L. Bianchi, Nanomaterials, Manuscript ID nanomaterials-1443545.
The authors replied satisfactorily to all of the questions and comments.
Minor comments:
- In Figure 3S, please indicate in the caption what image B refers to. Moreover, in image A, P/PU does not seem to have a CA = 74.4°, but a higher one (it seems comparable to image E), and, in image C, PT/PU-2 seems to be more hydrophobic than PT/PU-3 (image D), differently from what is indicated in Table 1.
Author Response
Please, see the attachment.
